# A systematic view on influenza induced host shutoff

Adi Bercovich-Kinori[1†], Julie Tai[1†], Idit Anna Gelbart[1], Alina Shitrit[1], Shani Ben-Moshe[2], Yaron Drori[3,4,5,6], Shalev Itzkovitz[2], Michal Mandelboim[3,4,5,6], Noam Stern-Ginossar[1*]

[1]Department of Molecular Genetics, Weizmann Institute of Science, Rehovot, Israel; [2]Department of Molecular Cell Biology, Weizmann Institute of Science, Rehovot, Israel; [3]Central Virology Laboratory, Chaim Sheba Medical Center, Ministry of Health, Rehovot, Israel; [4]Department of Epidemiology and Preventive Medicine, Tel-Aviv University, Tel-Aviv, Israel; [5]School of Public Health, Tel-Aviv University, Tel-Aviv, Israel; [6]Sackler Faculty of Medicine, Tel-Aviv University, Tel-Aviv, Israel

**Abstract** Host shutoff is a common strategy used by viruses to repress cellular mRNA translation and concomitantly allow the efficient translation of viral mRNAs. Here we use RNA-sequencing and ribosome profiling to explore the mechanisms that are being utilized by the Influenza A virus (IAV) to induce host shutoff. We show that viral transcripts are not preferentially translated and instead the decline in cellular protein synthesis is mediated by viral takeover on the mRNA pool. Our measurements also uncover strong variability in the levels of cellular transcripts reduction, revealing that short transcripts are less affected by IAV. Interestingly, these mRNAs that are refractory to IAV infection are enriched in cell maintenance processes such as oxidative phosphorylation. Furthermore, we show that the continuous oxidative phosphorylation activity is important for viral propagation. Our results advance our understanding of IAV-induced shutoff, and suggest a mechanism that facilitates the translation of genes with important housekeeping functions.

*For correspondence: noam.
stern-ginossar@weizmann.ac.il

[†]These authors contributed
equally to this work

Competing interests: The
authors declare that no
competing interests exist.

Reviewing editor: Nahum
Sonenberg, McGill University,
Canada

## Introduction

Influenza A viruses (IAV) are included among the Orthomyxoviridae family of negative single-stranded, segmented RNA viruses. These viruses cause an infectious disease that constitutes an important public health problem and remains today an important cause of morbidity and mortality (*Fields , 2007*). Like all viruses, IAV is absolutely dependent on the host-cell protein synthesis machinery to produce its proteins. To ensure priority access to host translation machinery, many viruses utilize host shutoff mechanisms that eliminate competition from cellular transcripts (*Walsh et al., 2012*). Host shutoff could be achieved by two complementary mechanisms: 1. Direct co-opting of the translation machinery by mechanisms that force better translation of viral mRNAs compared to their host counterparts. A classic example for this strategy is employed by poliovirus. It cleaves an essential host cap- binding protein, eIF4G, therefore preventing cap-dependent translation of host mRNAs while viral RNA translation stays unperturbed through the use of an internal ribosome entry site (*Ventoso et al., 1998*). 2. Viral-induced degradation of host mRNAs. This strategy is employed by several herpesviruses that express endonucleases which cleave host mRNAs, thereby eliminating the competition with host mRNA for the translation apparatus and ensuring efficient cap-dependent translation of viral mRNAs (*Glaunsinger and Ganem, 2006*).

IAV has long been known to significantly shutoff host gene expression. Interestingly, it is one of the few viruses for which both direct translation co-opting and host mRNA degradation were

**eLife digest** Proteins carry out diverse activities in our cells. These proteins are constantly being built according to accurate instructions, which are encoded on molecules named messenger RNAs (mRNAs for short). The process of converting the instructions into proteins is called translation.

Viruses infect host cells and take over the cellular machinery that is responsible for translation. This causes the cell to produce viral proteins at the expense of host proteins – a process called host shutoff. As a result, viral proteins take over the cell and the infection accelerates. There are two main strategies used by viruses to co-opt the cell's translation machinery: either host mRNAs are destroyed, or the machines that read mRNA molecules are manipulated to read only the viral instructions. Most viruses appear to dedicate themselves to using just one of these strategies. However, evidence suggests that the Influenza A virus uses both strategies to induce host shutoff.

To investigate the extent to which each of the shutoff strategies is used by the Influenza A virus, Bercovich-Kinori, Tai et al. have studied infected human lung cells. This revealed that the virus primarily reduces the amount of host mRNA in the cells to take over the mRNA pool. The host mRNAs were affected by the infection to different extents. For example, the mRNAs that coded for proteins that perform important roles for the virus, such as produce energy, were not affected by the virus.

A future challenge is to find out exactly how the Influenza A virus distinguishes between different cellular mRNAs. This knowledge may help to develop new treatments for flu.

suggested to play a prominent role in host shutoff (*Yanguez and Nieto, 2011*; *Jagger et al., 2012*). Although IAV mRNAs share the basic features with host mRNAs like a 5' 7-methyl guanosine (m7G) cap and a 3' poly-adenylate (poly(A)) tail, previous research suggested that influenza mRNAs are preferably translated due to features found in the 5'UTR of viral mRNAs (*Garfinkel and Katze, 1993*; *Katze et al., 1986*; *Park and Katze, 1995*). In addition host mRNA degradation has long been acknowledged (*Beloso et al., 1992*). The 5' m7G caps on viral transcripts are acquired by "cap-snatching" (*Plotch et al., 1981*), this snatching allows the priming of viral mRNA synthesis but also leads to nascent host transcripts degradation. In addition, inhibition of polyadenylation of host pre-mRNA by IAV Nonstructural protein 1 (NS1) protein (*Nemeroff et al., 1998*), and degradation of the host RNA polymerase II complex (*Rodriguez and Pérez-González, 2007*) could contribute to the reduction in host transcripts in infected cells. Recently, a highly conserved IAV protein PA-X, possessing the PA endonuclease domain, was shown to selectively degrade host mRNAs (*Jagger et al., 2012*), strongly suggesting that PA-X is a general influenza host shutoff endonuclease.

Although the discovery of PA-X has prompted new examination of the mechanisms that drive host shutoff during IAV infection (*Khaperskyy and McCormick, 2015*; *Khaperskyy et al., 2016, 2014*; *Bavagnoli et al., 2015*), several fundamental questions remain unanswered: (a) To what extent does IAV change host mRNAs expression and translation?, (b) what is the relative contribution of host mRNA degradation versus direct manipulation of translation to host shutoff? (c) to what extent does the virus possess mechanisms to ensure more effective translation of its own mRNAs?

Here, we took a systemic approach to explore the relative contribution of direct co-opting of the translation machinery, and reduction of host RNA levels to the reduction in host protein synthesis. To this end, we have used RNA sequencing (RNA-seq) and ribosome profiling (deep sequencing of ribosomes-protected fragments) to globally map the changes in host genes RNA and translation levels during IAV infection. These comprehensive and simultaneous measurements complemented with Single molecule Fluorescence in-situ Hybridization (smFISH) revealed that host shutoff is mainly achieved by reduction in cellular mRNAs levels, and that IAV transcripts are not preferably translated. Our systematic analysis also reveals that host transcripts are affected differently by IAV infection and that the extent of mRNAs reduction is related to their length and GC content. Interestingly, we noticed that transcripts encoding oxidative phosphorylation related proteins are less affected by IAV infection, their protein levels remain stable throughout infection and their continuous expression supports the energetic demands that are essential for virus replication.

## Results

### Simultaneous monitoring of RNA levels and translation during IAV infection

To gain a detailed view of the changes that occur in viral and host transcripts abundance and translation over the course of IAV infection, we infected A549 cells with the A/Puerto Rico/8/1934 H1N1 (PR8) strain at MOI = 5 and harvested cells at 2, 4, and 8 hr post infection (hpi). We designed our experiment to simultaneously monitor both RNA levels and translation (*Figure 1A*). Deep sequencing of mRNA (RNA-seq) allows a detailed mapping of transcript levels during infection and these were paired with ribosome footprints, which allow accurate measurement of protein synthesis by capturing the overall in vivo distribution of ribosomes on a given message (*Ingolia et al., 2011*). In order to assess the reproducibility of our experiments we prepared two independent biological replicates for each of these time points. Both the mRNA and footprints read density measurements were reproducible (*Figure 1B*). We quantitatively assessed the expression pattern of 7211 human transcripts and the 8 viral transcripts that are expressed from the 8 genomic segments of influenza. Metagene analysis, in which gene profiles are aligned and then averaged, revealed the expected profiles of footprints along mRNAs; ribosome density accumulates along the gene body ending at the first in-frame stop codon with pronounced accumulation of ribosomes at the initiation and

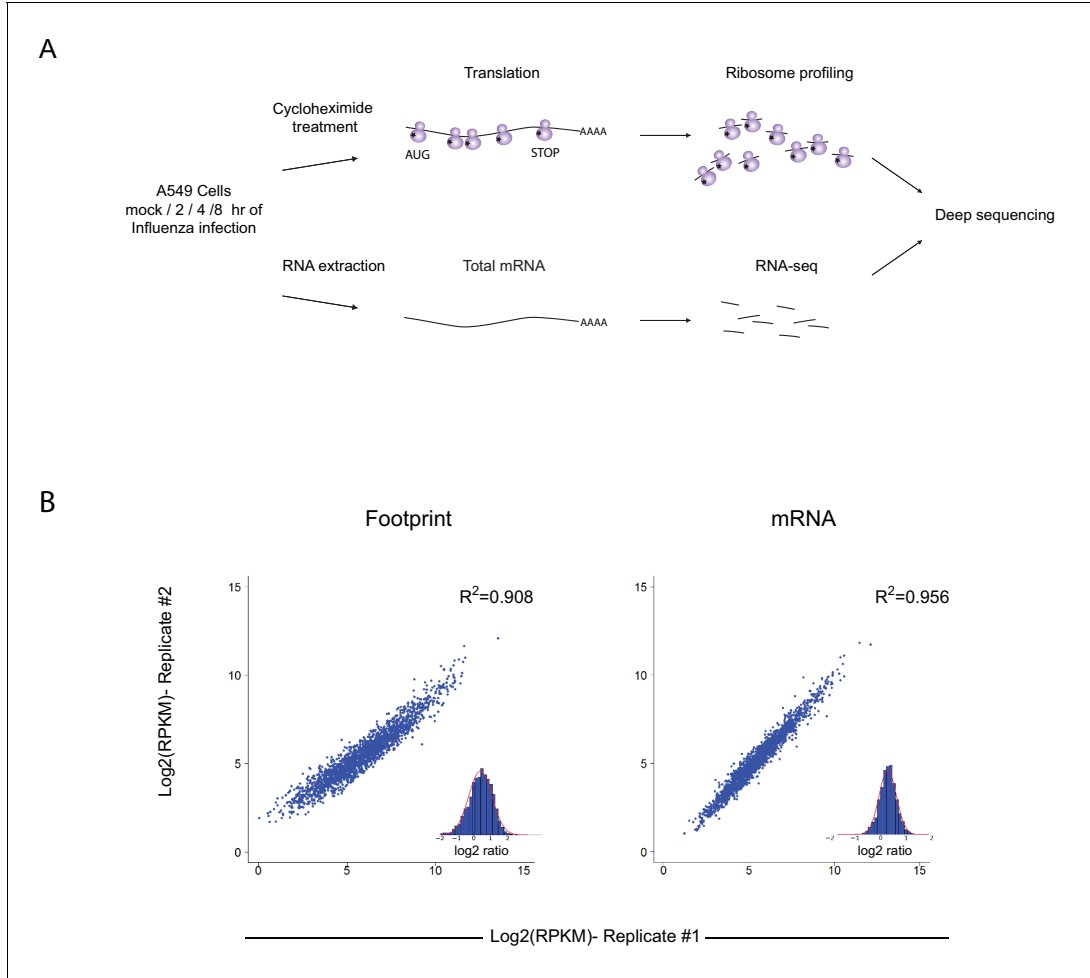

**Figure 1.** Ribosome profiling was performed along IAV infection. (**A**) Experiment set up of Ribosome profiling and RNA-seq along IAV infection. (**B**) Reproducibility of the ribosome footprints and mRNA measurements of host genes at 4 hpi. The correlation in footprints and mRNA measurements between biological replicates is presented.

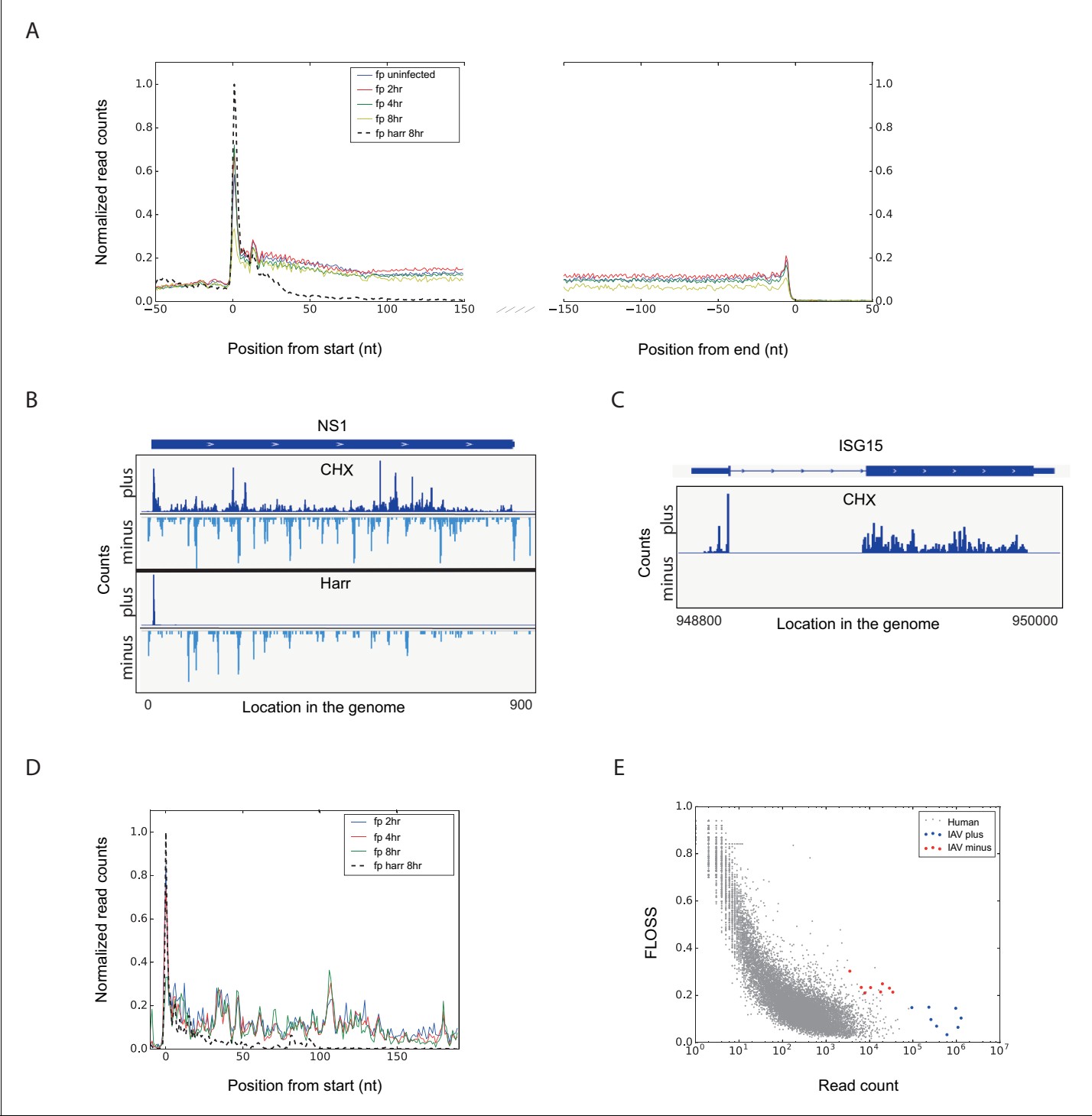

**Figure 2.** Footprint profiles reflect ribosome-protected fragments from host and viral mRNAs. (**A**) Metagene analysis of ribosome profiling data. Average ribosome read density profiles over well-expressed genes, aligned at their start codon and stop codons. Data from uninfected, 2, 4 and 8 hpi samples, and harringtonine pretreated 8 hpi sample. (**B**) Ribosome occupancies following treatments (illustrated on the left) with cycloheximide (CHX, top panels), and harringtonine (Harr, bottom panels) of the Influenza NS1 genomic locus at 8 hpi. Data from the plus strand (dark blue) and the minus strand (light blue) is presented. (**C**) Ribosome footprints profile of one human gene (ISG15) at 8 hpi. (**D**) Meta gene analysis from start codon for footprints originating from viral mRNAs. Data from 2, 4 and 8 hr of IAV infected and 8 hpi-harringtonine treated cells. (**E**) Fragment length organization similarity score (FLOSS) analysis for human mRNAs and for viral reads originating from the plus and minus strands.

*Figure 2 continued on next page*

*Figure 2 continued*

The following figure supplements are available for figure 2:

**Figure supplement 1.** Metagene analysis of ribosome profiling data originating from IAV minus strand.

**Figure supplement 2.** Percentage of footprints reads that aligned to IAV plus or minus strands from the sum of aligned viral reads at 2, 4, and 8 hpi.

termination sites (*Figure 2A*). Unexpectedly, examination of the ribosome profiling data obtained from influenza transcripts revealed reads that align to the IAV minus strand (vRNA) that is non-coding (*Figure 2B*). These reads were not correlated with any sequence feature related to translation and were specific for the virus as individual human transcripts presented the expected profiles precluding any general problem in the sample preparation (*Figure 2C*). To test if the footprints we obtained from viral mRNAs indeed originate from the ribosome protected fragments we generated an additional set of ribosome profiling libraries in which cells were pre-treated with harringtonine, a drug which leads to a strong accumulation of ribosomes precisely at translation initiation sites (*Ingolia et al., 2011*). As expected harringtonine treatment led to strong accumulation of ribosome protected fragments at the first AUG and to the depletion of ribosome density from the body of the viral and host mRNAs. Thus, this indicates that the protected fragments we captured from both cellular and viral mRNAs originate from ribosome protected fragments of transcripts that were engaged in active translation elongation (*Figure 2A,B and D*). In contrast, the protected fragments that mapped to the IAV minus strand were not affected by harringtonine treatment (*Figure 2B* and *Figure 2—figure supplement 1*) indicating that the protection of these fragments is probably not mediated by translating ribosomes.

To further illustrate that the footprints we obtained from viral mRNA reflect ribosome-protected fragments we applied a recently developed metric that distinguishes between 80S footprints and non-ribosomal sources using footprint size distributions (*Ingolia et al., 2014*). In ribosome-profiling data, the overall size distribution of fragments derived from protein-coding sequences, differs from the lengths of contaminating fragments found in profiling samples (*Ingolia et al., 2014*). We used a fragment length organization similarity score (FLOSS) that measures the magnitude of disagreement between the footprints distribution on a given transcript and the footprints distribution on canonical CDS. As expected thousands of well-expressed protein-coding transcripts scored well, and the similarity improved with increasing read counts ([*Ingolia et al., 2014*] and *Figure 2E*). IAV mRNAs scored well in these matrixes and they did not differ from well-expressed human transcripts (*Figure 2E*, blue). However, reads from IAV minus strand could be clearly distinguished from annotated coding sequences (*Figure 2E*, red). We conclude that the protected fragments originating from minus strand viral RNA are not generated by ribosome protection. These fragments could originate from protection by the viral nucleoprotein (NP) that associates with viral genomes and was shown to sediment with viral RNAs (*Duesberg, 1969*). This possibility is supported by the observation that the numbers of protected reads originating from the IAV genome increases as the infection progresses (*Figure 2—figure supplement 2*).

## IAV host shutoff is driven by reduced levels of host RNA but not reduced translation

In order to quantitatively evaluate if IAV evolved mechanisms to co-opt the cells' ribosomes we calculated the translation efficiency (TE) across our time course. TE is defined as the ratio of footprints to total mRNAs for a given gene and reflects how well a gene is being translated. We then compared the TE of human genes to that of viral genes at each of the time points along infection (*Figure 3A*). The analysis shows that the viral genes translation efficiencies fall within the general range of host gene translation indicating that viral transcripts are not preferentially translated during IAV infection and that direct co-opting of the translation machinery does not play a dominant role in cellular gene shutoff. Instead, our systematic analysis demonstrates that host shutoff stems from the viral dominance over the mRNA pool. At 8 hpi viral mRNAs take over 53.8% of the translation activity in the cells as 57.3% of the mRNAs in the cells are viral (*Figure 3B*). These findings show that the relative reduction in cellular mRNA expression is the main cause for host shutoff during infection.

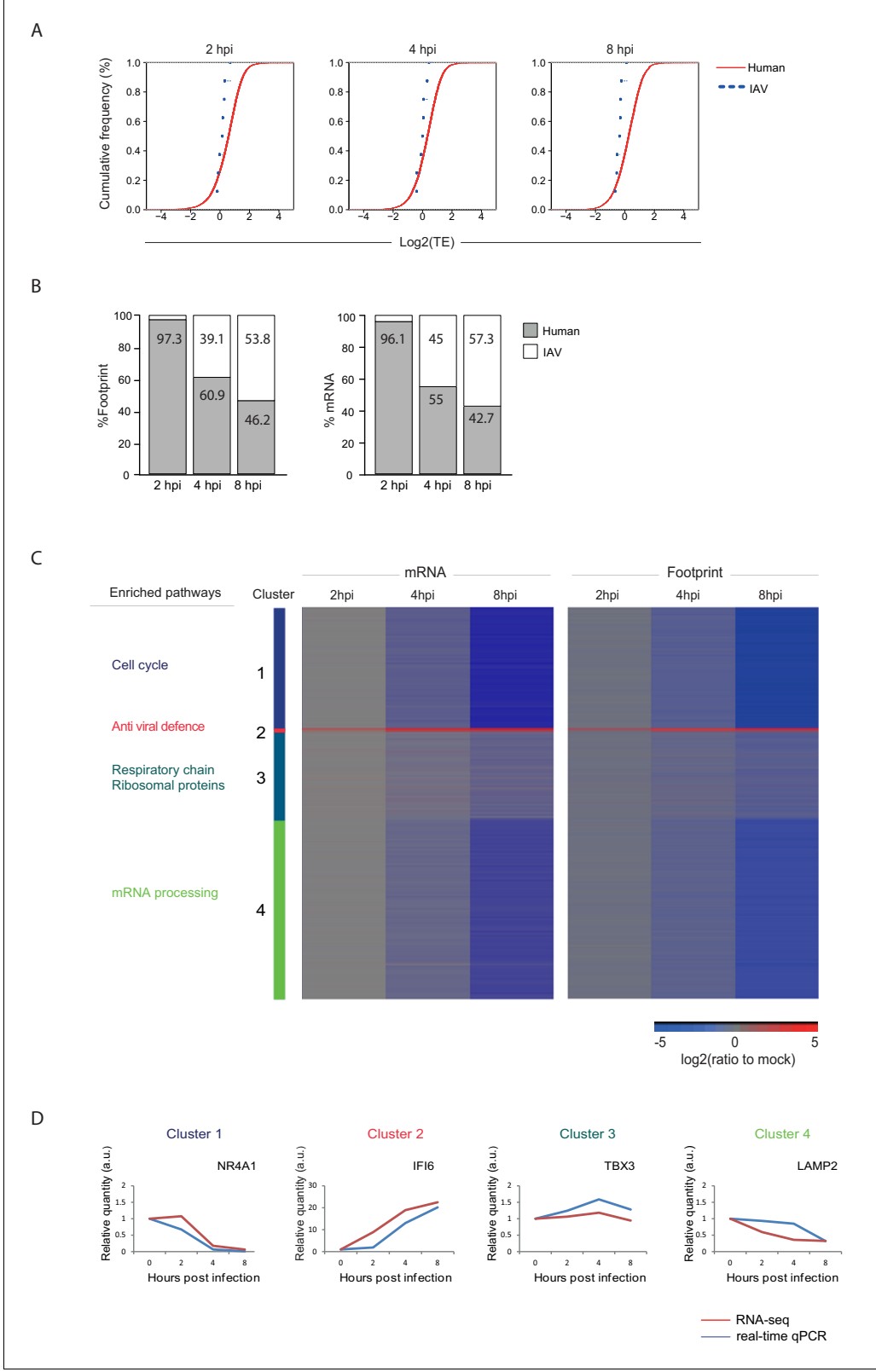

**Figure 3.** Reduction in cellular mRNA levels is driving the reduction in host protein synthesis. (**A**) Cumulative TE distribution among well-expressed human and viral genes shows that viral genes are translated at the same efficiency as cellular genes at 2, 4 and 8 hpi. (**B**) Percent of reads that aligned to the human or viral genome from the sum of aligned reads shown for ribosome profiling (footprints) and RNA-seq (mRNA) at 2, 4, and 8 hpi. (**C**) Ribosome footprints and mRNA read densities (reads per kilobase million, RPKM) of well-expressed human transcripts across three time points during

*Figure 3 continued on next page*

*Figure 3 continued*

Influenza infection were calculated relative to expression of uninfected cells (mock). Shown is a heat map of log2 expression ratios after partitioning clustering. Four main clusters are marked and for each of these clusters the pathway enrichment is labeled on the left. (D) Validation of mRNA measurements by real time qPCR for example genes. A representative analysis of two independent experiments is shown

The following figure supplement is available for figure 3:

**Figure supplement 1.** Lysates from A549 cells infected with IAV and metabolically labeled for 30-min periods ending at the times shown were analyzed by SDS-PAGE and autoradiography.

Our analysis concurs with the finding that PA-X endonuclease activity plays a dominant role in host shutoff during IAV infection (*Jagger et al., 2012*; *Khaperskyy and McCormick, 2015*; *Khaperskyy et al., 2016*, *2014*; *Bavagnoli et al., 2015*).

To confirm that our system indeed recapitulates the previous observations that IAV induces host shutoff (*Katze et al., 1986*; *Skehel, 1972*), we monitored protein synthesis in IAV infected A549 cells by metabolic labelling. As was previously reported, also in our system IAV infection produced abundant quantities of virus polypeptides from 4 hpi and this was accompanied by reduced cellular protein synthesis (*Figure 3—figure supplement 1*).

## Host transcripts are differently affected by IAV infection

Next we quantitatively assessed the expression pattern of cellular genes along IAV infection. Interestingly, 74% of host transcripts were reduced by more than three-fold in their footprints densities along infection, reflecting the drastic shutoff in host protein synthesis (*Supplementary file 1*). We compared the expression of the infected samples to mock sample and clustered the mRNA and footprint ratios using partitioning clustering. This approach allowed clustering of the cellular transcripts into 4 distinct classes based on similarities in temporal expression profiles in the RNA-seq and ribosome profiling data. Overall, we found that changes in ribosome footprints tracked the changes in transcripts abundance (*Figure 3C*), which is in line with host protein synthesis shutoff being driven by the in reduction cellular RNA levels and not by direct interference with the translation machinery. Interestingly, although the majority of host transcripts were significantly reduced during influenza infection, (cluster 1 and cluster 4, *Figure 3C*) we identified numerous genes that were significantly elevated (cluster 2, *Figure 3C*) and genes that were not significantly changed during influenza infection (cluster 3, *Figure 3C*).

We next carried out GO term enrichment analysis for each of these four clusters. As expected the small group of upregulated mRNAs was significantly enriched with genes related to antiviral defense (cluster 2, Pval = 6.9E-13), including Viperin and IDO1 that were previously shown to be elevated during IAV infection (*Ranaware et al., 2016*; *Wang et al., 2007*; *Huang et al., 2013*). Interestingly, cluster 1 which is composed of mRNAs that are most drastically downregulated is enriched with genes related to DNA repair and cell cycle (Pval = 1.7E-17 and Pval = 4.3E-18 respectively) whereas cellular mRNAs that are barely affected by IAV infection (cluster 3) are enriched with genes related to oxidative phosphorylation and for transcripts encoding ribosomal proteins (Pval = 3.4E-10 and Pval = 3.9E-8, respectively). These measurements and analysis reveal that the shutoff in host protein synthesis is mainly driven by reduction in cellular mRNA levels, and that host mRNAs sensitivity to influenza is not uniform and certain mRNAs are more resistant to viral interference.

One limitation of global deep-sequencing measurements is that these measurements provide relative but not absolute quantification of RNA and translation levels. To address this issue we confirmed that there are no significant changes in the overall levels of translation (*Figure 4B* and *Katze et al., 1986*) and in total RNA (*Supplementary file 2*) along IAV infection. We also extracted RNA from an equal number of cells along infection and performed real-time PCR analysis for cellular mRNAs from each of these clusters. These independent measurements were in strong agreement with our RNA-seq measurements (*Figure 3D*).

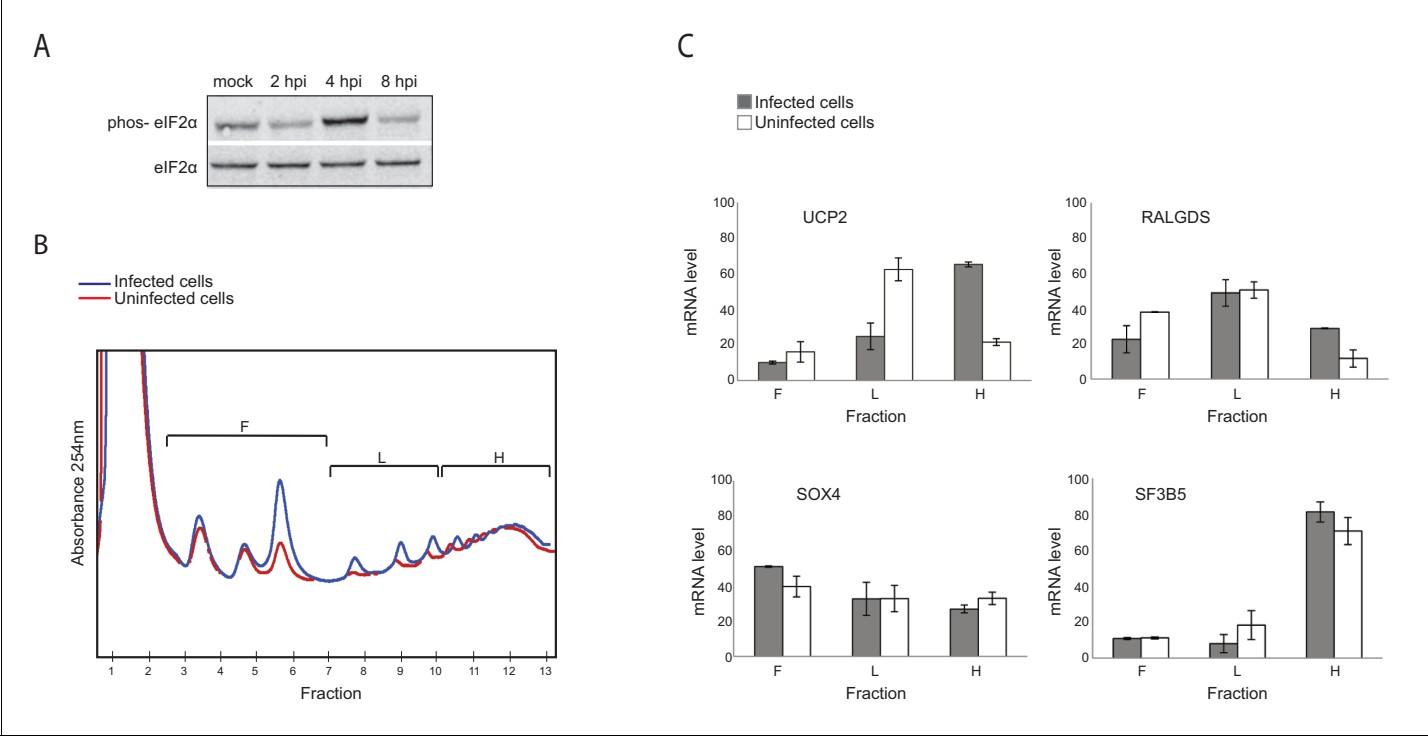

**Figure 4.** Genes responsive to eIF2α phosphorylation are translationally induced during IAV infection. (A) Western blot analysis of eIF2α and its phosphorylated form (phos- eIF2α) along IAV infection. A representative blot of two independent experiments is shown. (B) Polysomal profiling of uninfected and IAV infected cells at 4 hpi. (C) Real time qPCR analysis of the relative levels of the indicated human mRNAs in the polysome-free fractions of the gradient (F), light polysomal fractions (L), and heavy polysomal fractions (H) of uninfected and IAV infected cells at 4 hpi. A representative analysis of two independent experiments is shown

## Genes responsive to eIF2α phosphorylation are translationally induced after IAV infection

To quantitatively evaluate if there are genes that are differently translated along IAV infection, we used a computational framework named Babel (*Olshen et al., 2013*). This analysis quantifies levels of ribosome occupancy higher or lower than those predicted from transcript abundance. In total, this approach allowed us to identify 210 cellular mRNAs that are differentially translated following IAV infection (Pval<0.01, *Supplementary file 3*). Among these, 125 genes were translationally upregulated whereas 85 genes were translationally down regulated. While Go enrichment analysis did not reveal any significant functional category, we noticed that within the group of translationally upregulated genes at 4 hpi, there was an enrichment for genes that were previously shown as translationally induced in conditions in which eukaryotic initiation factor 2α (eIF2α) is phosphorylated (pval<0.002, [*Andreev et al., 2015*; *Sidrauski et al., 2015*]). eIF2α phosphorylation is a stress response that reduces overall protein synthesis while enhancing the translation of specific transcripts whose products support adaptive stress responses. By performing western blot analysis, we show that at 4 hpi eIF2α phosphorylation is apparent and only at 8 hpi it is drastically reduced (*Figure 4A*). By analyzing polysome profiles prepared from uninfected and IAV infected cells at 4 hpi, we confirmed that IAV infection indeed induce the translation of UCP2 and RALGDS (two genes that are also translationally induced following stress induced eIF2α phosphorylation [*Andreev et al., 2015*]), but not that of control genes SOX4 and SF3B5 for which no change in translation was observed (*Figure 4C* and *Supplementary file 3*). These results demonstrate that although IAV blocks eIF2α phosphorylation (*Goodman et al., 2007*), at 4 hpi eIF2α phosphorylation affects the cellular translation. In addition, these findings provide another confirmation for the accuracy of our translation measurements.

## smFISH measurements validate the RNA-seq quantification and reveal that cellular mRNA down regulation occurs in the nucleus

Our global measurements implied that host shutoff is driven by the high levels of IAV mRNAs that outnumber cellular transcripts, and that some cellular mRNAs are less affected by IAV. We wanted to preclude normalization issues and to validate these observations using an independent quantification method that provides absolute measurements. To this end, we used the smFISH technique that enables visualization of single mRNA molecules in fixed cells (*Raj et al., 2008*). This technique relies on the specific hybridization of short DNA libraries that are coupled to a fluorophore and are complementary to a specific target mRNA sequence. Binding of multiple probes to the same transcript yields a bright dot, indicative of a single mRNA molecule. This method has been used in various systems allowing accurate quantification of mRNA molecules per cell (*Itzkovitz, 2011*).

We quantified the expression of four human mRNAs, two mRNAs that are significantly reduced during influenza infection (*CHML* and *KIF18A*, cluster 1) and two mRNAs that were less affected by infection (*MYC* and *CDKN1B*, cluster 3). We also tested the expression of one viral transcript, Hemagglutinin (*HA*). We designed a panel of fluorescently labeled probes, each composed of 48 20-base oligos complementary to the coding sequences of each of these genes. We used A549 cells that were left uninfected or IAV infected and then fixed at 2, 4, and 8 hpi. Hybridization of these cells with the fluorescently labeled probes libraries yielded bright diffraction-limited dots, representing single transcripts. These were automatically counted using a custom image-processing software (*Bahar Halpern, 2016*). Importantly, when we examined *HA* expression in uninfected cells we did not detect any diffraction-limited dots demonstrating the specificity of this approach (*Figure 5A*). Following infection with IAV, *HA* transcript levels were drastically elevated and at 8 hpi the intensity of the smFISH signal was too high to measure single dots (*Figure 5A*). The expression levels of the four cellular transcripts we measured by smFISH were highly correlated with our RNA-seq measurements (*Figure 5A and B*). These absolute quantifications of viral and host transcripts along IAV infection strongly support the notion that host shutoff is mainly driven by differences in mRNA levels. Furthermore, this data illustrates the variability in the levels of cellular transcripts reduction during IAV infection, suggesting that IAV-mediated degradation might act differently on different cellular transcripts.

Multiple mechanisms for IAV interference with cellular RNA expression have been described (*Nemeroff et al., 1998*; *Fortes et al., 1994*; *Rodriguez et al., 2007*; *Plotch et al., 1981*), all of which involve interference with processes occurring in the nucleus. The discovery of the highly conserved RNA endonuclease, PA-X, implied for the existence of cytoplasmic degradation machinery, since viral RNAses from other viruses were proposed to act in the cytoplasm (*Abernathy, 2015*). However, recent work presented evidence that PA-X activity might be restricted to the nucleus (*Khaperskyy et al., 2016*). On top of absolute quantification, smFISH also provides spatial information about mRNA molecule distribution in the cell. This allowed us to test the levels of endogenous cellular transcripts in the nucleus and cytoplasm along influenza infection. We quantified the cytoplasmic and nuclear levels of *CHML* and *KIF8A*, which showed a drastic reduction in their levels, and of *MYC* that showed subtler but still a significant reduction (*Figure 5B*). Interestingly, the nuclear and cytoplasmic levels of *CHML, KIF18A* and *MYC* were downregulated to the same extent (*Figure 5C*). These results strongly suggest that interference in cellular RNA expression along IAV infection occurs mainly in the nucleus.

## Cellular transcripts reduction along IAV infection is correlated with transcripts' length and GC content

We noticed that cellular genes respond differently to IAV infection and can be divided based on the level of reduction they present during infection (*Figure 3C*, clusters 1, 3 and 4). Hence, we were interested in the features differentiating between these cellular mRNA groups. If IAV interference with cellular transcript expression occurs mainly in the nucleus and there is no selectivity in this process then the decline in mRNA levels should be correlated with the cytoplasmic half-lives of mRNAs. Using recent measurements of mRNA half-lives in A549 cells (*Maekawa et al., 2015*) we identified a significant enrichment in mRNAs with long half-lives in cluster 3, which includes genes that were only mildly affected by IAV infection (*Figure 5D*, Pval = 0.005), but this cluster contained also many genes with short half-life. These results suggest that there are additional features that govern the

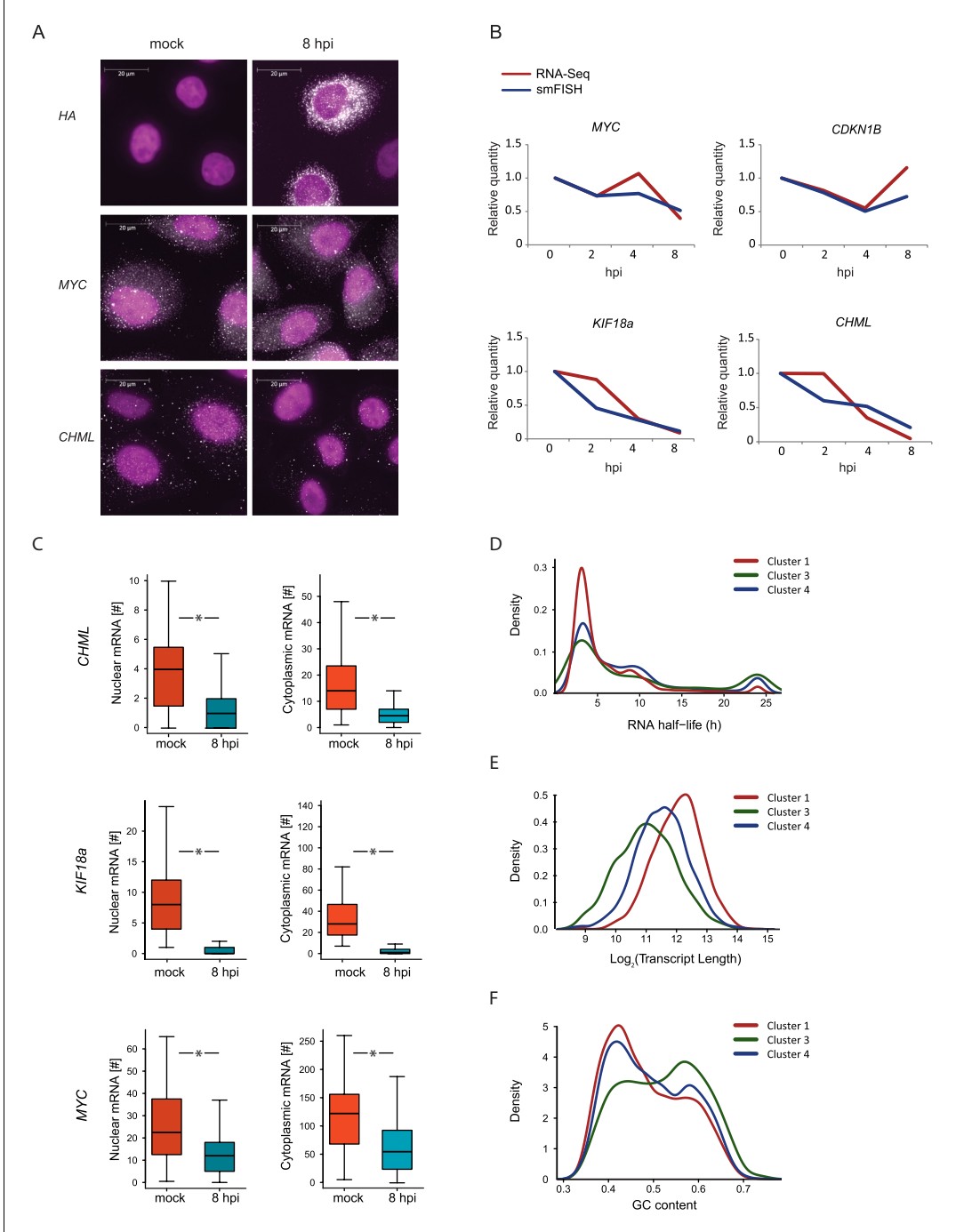

**Figure 5.** smFISH measurments of cellular mRNAs along IAV infection. (**A**) mRNA detection by Single molecule FISH (smFISH) was performed on A549 cells, either mock infected or 8 hpi. DNA oligomer probes coupled with fluorescent dyes (cy5/Alexa549) were targeted against viral hemagglutinin (HA), human MYC and CHML mRNAs. Spots corresponding to single mRNA molecules were detectable. DAPI was used for nuclear staining. Scale bar is 20 μm. Representative images of at least two biological replicates are shown. (**B**) The numbers of mRNA molecules per cell from at least 45 cells were quantified along Influenza infection (mock infection, 2, 4, 8 hpi) using smFISH. The values obtained from both smFISH and RNA-seq measurements were normalized to mock and plotted on the same graph. (**C**) For *CHML, KIF18a* and *MYC* nuclear and cytoplasmic mRNA molecules were quantified using smFISH. p-values are derived from a Student's *t*-test, *p<0.05. (**D–F**) Various features were compared between transcripts composing the clusters presented in Fig. 3C (representing differential response to IAV infection). (**D**) mRNA half-lives measurements in A549 cells (33) (**E**) Transcripts length distribution. (**F**) Transcripts GC content distribution.

The following figure supplements are available for figure 5:

*Figure 5 continued on next page*

*Figure 5 continued*

**Figure supplement 1.** Poly-A tail length of host transcripts as was previously measured from HeLa and NIH3T3 cells (*Chang et al., 2014*) were compared between transcripts composing the clusters presented in *Figure 3C*.

**Figure supplement 2.** Transcript's GC content (left) and length distribution (right) were calculated for genes encoding ribosomal proteins and respiration components, for genes in cluster 3 after omitting these functional categories (cluster 3*) and for clusters 1 and 4 presented in *Figure 3C* (representing differential response to IAV infection).

differences between these clusters, and that the differences in the levels of reduction might stem from differences in IAV-interference with host genes expression.

Since our measurements suggested that most of the IAV-mediated reduction occurs in the nucleus and a recent study connected PA-X activity to the 3' end processing (*Khaperskyy et al., 2016*), we tested whether the length of the poly-A tail affects the extent to which mRNAs are reduced after IAV infection. Using genome wide measurements of poly-A tail length (*Chang et al., 2014*) we did not observe any significant differences between the different clusters (*Figure 5—figure supplement 1*). We next examined specific characteristics of the corresponding transcripts, including their length and GC content. Interestingly, both mRNA length and GC content showed a significant difference between the clusters, and the transcripts that were less affected by IAV were significantly shorter and had higher GC content (*Figure 5E and 5F*, Pval≤1.49e-63 and Pval≤3.636e-06, respectively). Since cluster 3 (composed of genes that were less affected by IAV) is also significantly enriched in transcripts related to oxidative phosphorylation and ribosomal proteins, and these functional categories are composed of genes that tend to be short (*Figure 5—figure supplement 2*), we wanted to exclude the possibility that the reduced length we observed is indirectly driven by the enrichment in these functional categories. Indeed, also after omitting transcripts related to these functional categories cluster 3 transcripts were significantly shorter (*Figure 5—figure supplement 2*). Taken together, these results strongly suggest that shorter and more structured transcripts (transcripts with higher GC content) tend to be less affected by IAV infection and therefore it is likely that the extent to which cellular mRNAs are degraded is at least partially governed by the quantity of exposed single stranded RNA in the nucleus.

## The oxidative phosphorylation capacity is maintained late in IAV infection

Host shutoff mechanisms clearly provide advantages for viruses as they hamper the cellular response to infection and limit competition for cellular ribosomes. However, dampening cellular gene expression also poses a big challenge, as viruses are fully dependent on cellular resources such as macromolecules' building blocks, energy production and the host translational machinery. Therefore, the observation that transcripts coding for ribosomal and oxidative phosphorylation proteins, are less reduced following IAV infection, implies that the translation maintenance of these cellular pathways might be important for IAV infection. To validate this notion we conducted western blot analysis on cell extracts along IAV infection and could confirm the stable expression of components of the respiratory complex (*Figure 6A*, upper panel), whereas other proteins were reduced following IAV infection (*Figure 6A*, lower panel). In accordance with these results, by staining infected cells with the potential sensitive mitochondrial stain Tetramethylrhodamine (TMRM), we show that the mitochondria membrane potential is not affected by IAV infection (*Figure 6B*). Next we tested if continuous oxidative phosphorylation is important for IAV propagation by employing drugs that interfere with the function of the respiratory chain. We used either the ionophore Carbonyl cyanide m-chlorophenyl hydrazone (CCCP) or Valinomycin that chelates potassium ions, both causing a collapses of the mitochondria membrane potential. When cells were infected for 2 hr and then treated with CCCP or Valinomycin in concentrations that did not affect cell viability (*Figure 6—figure supplement 1*), IAV titers were significantly reduced (*Figure 6C and D*). Overall, these results indicate that mRNAs involved in oxidative phosphorylation are less perturbed by IAV infection, and that maintenance of intact oxidative phosphorylation activity during IAV infection is important for viral replication.

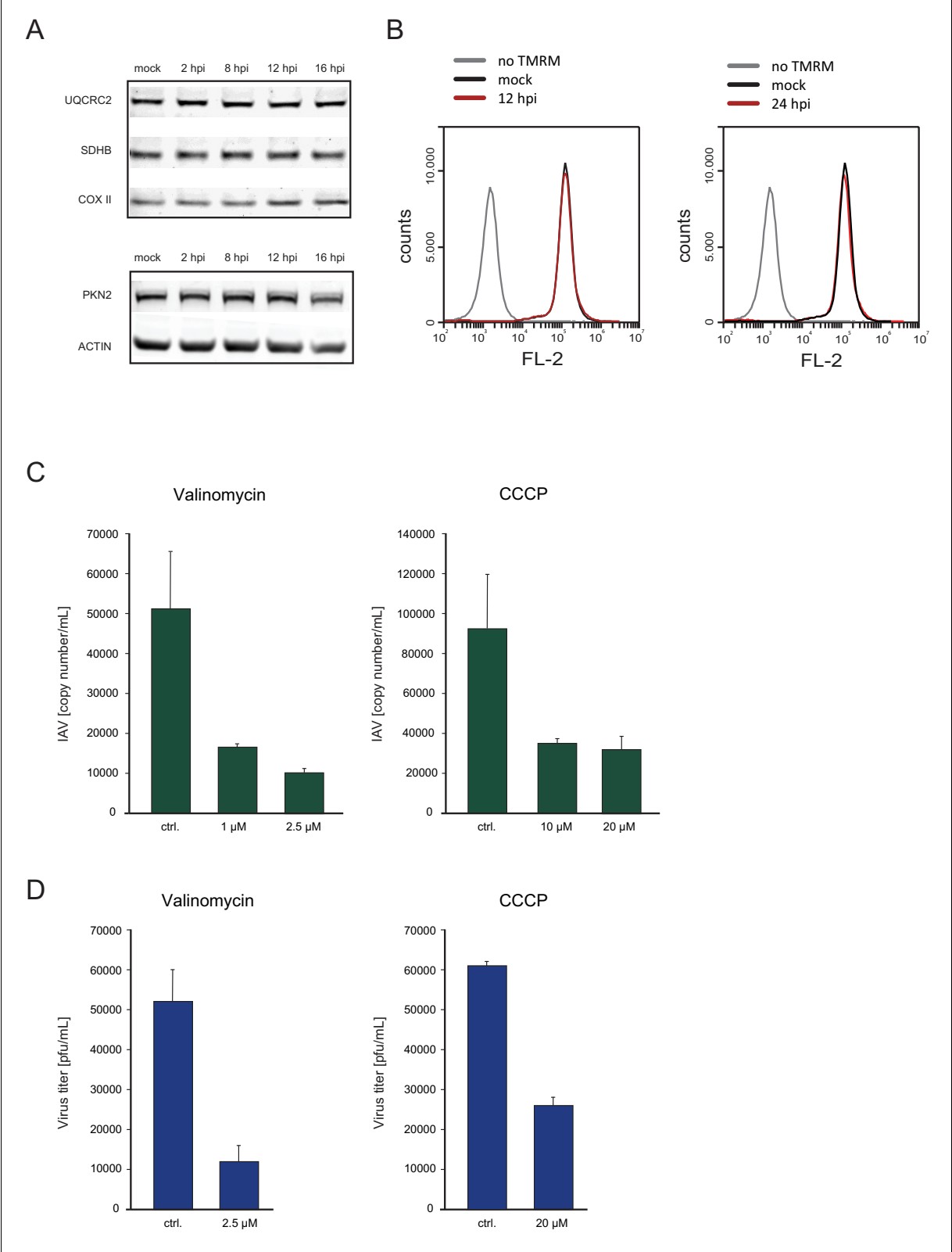

**Figure 6.** Oxidative phosphorylation capacity is important for IAV replication. (**A**) Western blot analysis of components of the respiratory complex UQCRC2, SDHB, COXII, and control genes PKN2 and ACTIN along IAV infection. A representative blot of three independent experiments is shown. (**B**) FACS analysis of TMRM staining for active mitochondria in A549 cells, either mock infected, untreated, infected for 12 and (left panel) or 24 hr (right panel). A representative analysis of two independent experiments is shown. (**C** and **D**) A549 cells infected with IAV were treated with either CCCP
*Figure 6 continued on next page*

*Figure 6 continued*

(untreated control, 10 μM and 20 μM) or Valinomycin (untreated control, 1 μM and 2.5 μM). Supernatants were collected at 24 and 48 hpi. (**C**) Viral copy numbers were estimated by qPCR and (**D**) viral titers were measured by plaque assay. Representative results of two independent experiments are shown.

The following figure supplement is available for figure 6:

**Figure supplement 1.** FACS analysis of propidium iodide (PI) staining of A549 cells infected with IAV for 48 hr and treated with either CCCP (untreated, 10 μM or 20 μM) or Valinomycin (untreated, 1 μM, 2.5 μM).

## Discussion

Many viruses have developed varied and sophisticated mechanisms to specifically repress cellular mRNA translation and concomitantly allow the selective translation of viral mRNA. In the case of IAV many mechanisms were suggested to contribute to the observed host shutoff. These include: 1. Cap-snatching of cellular pre-mRNAs (*Plotch et al., 1981*), 2. Inhibition of cellular pre-mRNAs polyadenylation (*Nemeroff et al., 1998*), 3. Degradation of RNA Polymerase II (*Rodriguez and Pérez-González, 2007*), 4. Nuclear retention of cellular mRNAs (*Fortes and Beloso, 1994*), 5. Cellular mRNAs degradation (*Khaperskyy et al., 2016*; *Beloso et al., 1992*; *Inglis, 1982*) and 6. Preferential translation of viral mRNAs (*Yanguez and So similar, 2011*; *Garfinkel and Katze, 1993*; *Katze et al., 1986*). Our systematic and unbiased analysis surprisingly demonstrates that viral mRNAs are not preferentially translated compared to their host counterparts and that the extensive translation of viral proteins is the result of viral takeover of the mRNA pool in the cell. These results are consistent with the notion that NS1 is not required for the shutoff of host cell protein production (*Salvatore et al., 2002*) and that PA-X endonuclease activity is a dominant factor in mediating host shutoff activity (*Jagger et al., 2012*; *Khaperskyy and McCormick , 2015*; *Khaperskyy et al., 2016*, *2014*; *Bavagnoli et al., 2015*).

It should be noted that at 8 hpi there is already extensive viral replication (evident from the strong increase in reads originating from the viral minus strand). Therefore, at this time point a portion of the viral RNA plus strand we measured are used as a template for replication (cRNA) and not as mRNAs. Since our calculation of viral transcripts TE is based on dividing the number of footprints by the number of corresponding RNA reads it is possible that we underestimate the TE of viral genes at this time point. However, using our smFISH measurements for viral transcript we could show that >90% of the plus strand viral RNA is located in the cytoplasm. Therefore, this potential bias could result in a maximum of 10% effect that does not hamper our conclusions.

Previous studies demonstrated inhibition of eIF2a phosphorylation during influenza infection and it was proposed that this inhibition is mediated by the activation of the cellular inhibitor of PKR, P58-IPK (*Goodman et al., 2007*). Our analysis supports the observation that IAV infection inhibit eIF2a phosphorylation but we show that this inhibition is not immediate. In our experimental system eIF2a was phosphorylated at 4 hpi and this stress response affects the expression of cellular genes (*Figure 4*). Since our measurements suggest the P58-IPK (DNAJC3) expression is reduced during IAV infection (by three-fold) it is possible that other mechanisms also contribute to the inhibition of eIF2a phosphorylation.

Due to library normalization, it is challenging to assess from the ribosome profiling and RNA-Seq measurements on overall virus-induced host shutoff. To address this inherent limitation, we used smFISH, which allows absolute quantification of transcript levels in single cells. Our smFISH measurements correlated well with our global RNA quantification confirming our data and findings. In addition, our smFISH measurements show that after IAV infection RNA levels of endogenous cellular transcripts are reduced to a similar extent in the nucleus and the cytoplasm. This observation suggests that cellular mRNAs degradation occurs in the nucleus but it could also be explained by blockage of transcription in conjunction to cytoplasmic degradation that reduces cytoplasmic mRNA levels. The recent observation that PA-X activity is mostly nuclear (*Khaperskyy et al., 2016*) together with limited correlation with a cytoplasmic half-life strongly points to the former possibility.

Our analysis also allowed us to quantify the reduction in cellular mRNAs along IAV infection, revealing that the extent of reduction varies between different transcripts. The magnitude of

reduction could only be partially explained by cytoplasmic half-life and interestingly the extent of reduction was significantly correlated with transcript's length and GC content. Since cap-snatching activity was shown to target mostly small nuclear RNAs (*Koppstein et al., 2015*), it is likely that the majority of the observed differential reduction in cellular mRNAs is driven by PA-X degradation activity. Therefore, the relation to the length and GC content strongly points to a non-selective process of degradation, where the chances of a transcript to be degraded are influenced by the quantity of exposed ssRNA. This notion is consistent with existing in-vitro data showing PA-X preference to ssRNA (*Bavagnoli et al., 2015*) and with the findings that PA-X lacks obvious sequence or location specificity (*Khaperskyy et al., 2016*). However, the notion that IAV induced degradation significantly depends on the length of the transcript does not easily reconcile with the evidence that PA-X specifically targets Pol II transcripts and that this specificity is connected to 3' end processing (*Khaperskyy et al., 2016*). One way these results could be rationalized is if Pol II transcription and processing is physically limited to specific sub-nuclear compartments (*Mitchell, 2008*; *Ghamari et al., 2013*). In this case, the specificity of PA-X to Pol II transcripts could originate from physical proximity, while the recognition itself could be random and depends on the length and GC content, which will govern the amount of exposed ssRNA. Another possibility is that degradation activity is coupled to mRNA splicing. This can explain the specificity to Pol II transcripts and the relative resistance of short transcripts which intrinsically also tend to have less exons. However, this possibility seems less favorable as at least by over expression PA-X did not affect the expression of cellular nascent transcripts (*Khaperskyy et al., 2016*). Future work focused on PA-X activity can help shed light on the exact mechanism.

Host shutoff is a common phenomenon, employed by diverse viruses and it is thought to contribute to the viral progression in two main ways: by redirecting the translation machinery towards viral gene expression and by inhibiting cellular anti-viral responses. However, shutting off cellular protein production could also have adverse effects. Viruses are completely dependent on cellular resources and inhibiting cellular protein production can dampen processes that are indispensable for the virus. Interestingly, we noticed that transcripts that are less affected by IAV infection are enriched for genes involved in pathways that the virus potentially depends on, such as oxidative phosphorylation components and ribosomal proteins. In contrast, genes that were most significantly reduced are related to cell cycle, a cellular activity that is often blocked by viruses. Indeed, we were able to show that the levels of mitochondrial proteins are not affected by IAV even at 16 hpi, a time point in which other proteins were reduced. Further experiments demonstrated that inhibition of oxidative phosphorylation to levels that do not affect cell's viability severely impairs viral replication. These results suggest that continuous expression of proteins involved in oxidative phosphorylation is important for viral propagation. Since viruses mostly interfere with protein synthesis, the level of protein reduction during infection is also tightly connected to the half-life of these proteins. In this regard, the observation that many of the proteins that perform basic cellular functions like translation and cellular respiration have long half-lives (*Schwanhausser et al., 2011*) could also assist viruses to avoid detrimental damage from shutting-off cellular protein synthesis. The inherent difficulties in blocking host protein production should be common to other viruses inducing host shutoff, particularly viruses with long infection cycle. It will therefore be interesting to see whether other viruses that interfere with cellular protein synthesis will also do this in a differential manner, facilitating the maintenance of cellular functions that are important for viral propagation.

It is well acknowledged that ribosome profiling is an emerging technique that allows probing of translation systematically and with increased sensitivity. This method in conjunction with RNA-seq was previously used to map the changes that occur in host gene expression during herpes simplex virus and cytomegalovirus infections (*Rutkowski et al., 2015*; *Tirosh et al., 2015*) and to probe the complexity of several viruses (*Stern-Ginossar et al., 2012*; *Arias et al., 2014*; *Yang et al., 2015*; *Irigoyen et al., 2016*). Here we show how these methods in conjunction with smFISH could be used to accurately quantify host shutoff along viral infection. These unbiased and systematic quantification methods can advance our understanding of viral interference with host protein production and are applicable to infection with any virus.

# Materials and methods

## Cells and viruses

The cells used in this study were the human lung adenocarcinoma epithelial cell line A549. The human influenza viruses A/Puerto Rico/8/34 H1N1 used in this study were generated as previously described (*Achdout et al., 2003*). Cells were grown on 10 cm2 plates and were infected at a multiplicity of infection (MOI) of 5. Efficiency of infection was determined by immunofluorescence with Influenza A antibody FITC Reagent (LIGHT DIAGNOSTIC) followed by DAPI dye for nuclear staining, and confirmed that >95% of cells were infected.

Viral genomic RNA was quantified by extracting RNA from the supernatant by using the NucliSENS easyMAG (BioMerieux, France). Detection of influenza A virus infection was performed by real-time reverse transcription-PCR (rRT-PCR),using TaqMan Chemistry on the ABI 7500 instrument as previously described (*Hindiyeh et al., 2005*).

## Ribosome profiling and RNA-Seq samples preparation

Cycloheximide treatments were carried out as previously described (*Stern-Ginossar et al., 2012*). Cells were lysed in lysis buffer (20 mM Tris 7.5, 150 mM NaCl, 5 mM MgCl2, 1 mM dithiothreitol, 8% glycerol) supplemented with 0.5% triton, 30 U/ml Turbo DNase (Ambion) and 100 µg/ml cycloheximide, ribosome protected fragments were then generated as previously described (*Stern-Ginossar et al., 2012*). Total RNA was isolated from infected cells using Tri-Reagent (Sigma). Polyadenylated RNA was purified from total RNA sample using an Oligotex mRNA mini kit (Qiagen). The resulting mRNA was modestly fragmented by partial hydrolysis in bicarbonate buffer so that the average size of fragments would be ~80 bp. The fragmented mRNA was separated by denaturing PAGE and fragments 50–80 nt were selected and sequencing libraries were made as previously described (*Stern-Ginossar et al., 2012*).

## Sequence alignments, normalization and clustering

Prior to alignment, linker and polyA sequences were removed from the 3′ ends of reads. Bowtie v0.12.7 (allowing up to 2 mismatches) was used to perform the alignments. First, reads that aligned to human rRNA sequences were discarded. All remaining reads were aligned to the concatenated viral (EF467817 - EF467824) and human (hg19) genomes. Finally, still-unaligned reads were aligned to the human known canonical transcriptome that includes splice junctions. Reads with unique alignments were used to compute the total number of reads at each position. Footprints and mRNA densities were calculated in units of reads per kilobase per million (RPKM) in order to normalize for gene length and total reads per sequencing run.

The expression patterns were examined for genes that had more than 150 uniquely aligned reads of mRNA and footprints. Partitioning clustering was performed using Partek Genomic suits across mRNA and footprint data. GO enrichment analysis was done using DAVID database (*Huang et al., 2009*). The Babel computational framework was used to quantitatively evaluate if there are genes that are differently translated along the IAV infection (*Olshen et al., 2013*).

## $^{35}$S-labelling of nascent proteins

Infected A549 cells (mock, 2 hpi, 4 hpi, 8 hpi) were incubated in methionine depleted DMEM for 30 min. Subsequently, 50 µCi/mL [$^{35}$S]-methionine was added and incubated for 30 min. Cells were harvested using RIPA buffer. Lysates were run on a polyacrylamide (PAA) gradient gel (4–12%). The PAA gel was fixed overnight in a fixation solution (3% glycerol, 20% methanol, 7% acetic acid). Typhoon FLA 7000 from GE Healthcare Life Sciences was used for acquisition.

## Polysome profiling

Four hours post infection A549 cells were treated with 100 µg/ml Cycloheximide for 1 min and then washed twice with cold PBS containing 100 µg/ml Cycloheximide. The cells were collected and lyzed with 400 µl lysis buffer (20 mM Tris 7.5, 150 mM NaCl, 5 mM MgCl2, 1mM dithiothereitol) supplemented with 0.5% triton, 30 U/ml Turbo DNase (Ambion) and 100 µg/ml Cycloheximide. The lyzed samples were centrifuged at 12,000 g at 4°C for 10 min. The cleared lysates were loaded onto 10–50% sucrose gradient and centrifuged at 35,000 RPM in a SW41 rotor for 3 hr at 4°C. Gradients

were fractionated and the optical density at 254 nm was continuously recorded using Biocomp gradient station.

## Real-time PCR

RNA was isolated using Tri-Reagent (Sigma) and Direc-Zol RNA mini-prep kits (Zymo Research). cDNA was prepared from 1µg RNA except for polysomes profiling experiments in which 255 ng RNA was used with High-Capacity cDNA Reverse Transcription Kit (ABI). Real time PCR was performed using the SYBR Green PCR master-mix (ABI) on a real-time PCR system StepOnePlus (life technologies) with the following primers:

| Gene | Forward primer | Reverse primer |
| --- | --- | --- |
| TBX3 | CGAAGAAGAGGTGGAGGACG | AAACATTCGCCTTCCCGACT |
| UCP2 | AGCCCACGGATGTGGTAAAG | CTCTCGGGCAATGGTCTTGT |
| RALGDS | GGTAGATTGCCAGAGCTCCA | CCTTGTTGCCTCCGTGGT |
| SOX4 | GCACTAGGACGTCTGCCTTT | ACACGGCATATTGCACAGGA |
| SF3B5 | GCGACTCGTACTGCTCCTAC | CTCGCGCTTTGCTCTCATTC |
| LAMP2 | CCGGCTTCTGGAGTAAGGTA | CAGGTGTACAAAGCAGCCAT |
| IFI6 | TACACTGCAGCCTCCAACTC | AGTTCTGGATTCTGGGCATC |
| NR4A1 | GTCCTGGGTCCAGTAGGAAA | GAGGAGTGGGACTGACCAAT |

## smFISH

Probe library constructions, hybridization procedures, and imaging conditions were described previously (*Itzkovitz, 2011*). In short, probe libraries consisted of 48 probes of length 20 bases, were coupled to cy5 (for *MYC*, and KIF18A) or Alexa594 (for *CDKN1B, HA* and *CHML*). Hybridizations were performed overnight in 30°C. DAPI dye for nuclear staining was added during the washes. Images were taken with a Nikon Ti-E inverted fluorescence microscope equipped with a ×100 oilimmersion objective and a Photometrics Pixis 1024 CCD camera using MetaMorph software (Molecular Devices, Downington, PA). Quantification was done on stacks of 10–15 optical sections, with Z spacing of 0.2 µm.

## Fragment length organization similarity score (FLOSS)

The FLOSS score was computed as previously described (*Ingolia et al., 2014*). Briefly, a histogram of read lengths for all footprints that aligned to a specific transcript or reading frame was calculated and compared to a reference histogram produced by summing individual raw counts (without normalization) for annotated nuclear transcripts.

## Examination of various features of cellular transcripts

The transcripts composing the clusters presented in *Figure 3C* were analyzed. For each cluster, we examined recently published data of RNA half-life in A549 cells (*Maekawa et al., 2015*). In addition, we examined the mean poly(A) tail length from measurements that were performed in HeLa and NIH3T3 cells (*Chang et al., 2014*). Transcripts length and GC content were calculated excluding introns. For each feature, pairs of clusters were compared using a two-sided t-test except for RNA half-life in which the clusters were compared using the Mann-Whitney test. Presented is the largest p-value of the three comparisons.

## Western blot analysis and drug treatments

Infected A549 cells and the mock control were harvested 2 hpi, 4 hpi 8 hpi, 12 hpi and 16 hpi using RIPA Buffer. The membrane was blocked for 1 hr in 5% skim milk TBST. Primary antibodies UQCRC2, SDHB, COXII (ab110413, Abcam), PKN2 (ab32395, Abcam), ACTIN (A4700, Sigma Aldrich), eIF2$\alpha$ (sc-11386, Santa Cruz) and phosphorylated (S52) eIF2$\alpha$ (447286, Invitrogen) were diluted 1:1000 in 5% BSA TBST and incubated for 1 hr at RT or over night at 4°C. Secondary antibodies IRDye 680RD

goat anti-rabbit (LIC-92668071) and IRDye 680RD goat anti-mouse (LIC-92668070) were diluted 1:10000 in 5% skim milk TBST and incubated for 1 hr at RT. Acquisition was performed using, Odyssey CLx from LI-COR. A549 cells were treated after 2 hpi (IAV) with CCCP (untreated, 10 µM or 20 µM) or Valinomycin (untreated, 1 µM, 2.5 µM). For Tetramethylrhodamine (TMRM) staining A549 cells were incubated with 20 nM TMRM in 1ml of DMEM for 30 min at 37°C, prior analysis in the flow cytometer. For PI (propidium iodide) staining, target cells were resuspended in 100 µL PBS and 10 µL staining solution (10 µg/mL PI in PBS) was added for 1 min prior to acquisition in the flow cytometer.

## Plaque assay

Confluent MDCK cells were incubated for 1 hr at 37°C with 0.5 ml 10-fold serial dilutions of virus in 6-well plates, present in DMEM medium supplemented with 3 µg/mL IX-S trypsin (sigma T-0303). The cells were then washed and overlaid with freshly prepared DMEM containing 1.5% agarose. The plaques were visualized after incubation at 37°C for 3 days by staining with 0.3% crystal violet solution containing 20% ethanol.

## Acknowledgements

We thank Yossef Shaul, Nina Reuven, Ofer Mandelboim and NS-G lab members for critical reading of the manuscript. This research was supported by a research grant from Mr. Ilan Gluzman, The Human Frontiers Science Program Career Development Award, the EU-FP7-PEOPLE Career integration grant, the ICORE (Chromatin and RNA Gene Regulation), the Israeli Science Foundation (1073/14) and the European Research Council starting grant (StG-2014-638142). NS-G is incumbent of the skirball career development chair in new scientist.

## Additional information

### Funding

| Funder | Grant reference number | Author |
|---|---|---|
| Human Frontier Science Program | Career Development Award | Noam Stern-Ginossar |
| Israel Science Foundation | 1073/14 | Noam Stern-Ginossar |
| Israeli Centers for Research Excellence | 1796/12 | Noam Stern-Ginossar |
| European Research Council | Starting grant, StG-2014-638142 | Noam Stern-Ginossar |
| European Commission | EU-FP7-PEOPLE | Noam Stern-Ginossar |
| Mr. Ilan Gluzman | | Noam Stern-Ginossar |

The funders had no role in study design, data collection and interpretation, or the decision to submit the work for publication.

### Author contributions

AB-K, JT, Conception and design, Acquisition of data, Drafting or revising the article; IAG, Conception and design, Analysis and interpretation of data, Drafting or revising the article; AS, SI, Analysis and interpretation of data, Drafting or revising the article; SB-M, Acquisition of data, Analysis and interpretation of data; YD, Contributed unpublished essential data or reagents; MM, Drafting or revising the article, Contributed unpublished essential data or reagents; NS-G, Conception and design, Drafting or revising the article

### Author ORCIDs

Shalev Itzkovitz, http://orcid.org/0000-0003-0685-2522
Noam Stern-Ginossar, http://orcid.org/0000-0003-3583-5932

## Additional files

### Supplementary files

• Supplementary file 1. This table presents the fold change in RPKM of human genes RNA levels (mRNA) and translation (footprints) compare to the mock sample. The numbers are an average of two independent biological repeats.

• Supplementary file 2. This table lists the concentrations of total RNA (ng/µl) extracted along IAV infection in two independent biological repeats.

• Supplementary file 3. This table provides the statistical significance value (p-value) of translationally regulated genes (p val<0.01).  Positive values represent upregulation and negative values represent downregulation at the indicated time post infection.

### Major datasets

The following dataset was generated:

| Author(s) | Year | Dataset title | Dataset URL | Database, license, and accessibility information |
|---|---|---|---|---|
| Gelbart IA, Stern-Ginossar N | 2016 | A systematic view on Influenza induced host shut-off | www.ncbi.nlm.nih.gov/geo/query/acc.cgi?acc=GSE82232 | Publicly available at the NCBI Gene Expression Omnibus (accession no: GSE82232) |

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
