## [Decision Letter]

Thank you for submitting your article "A systematic view on Influenza induced host shutoff" for consideration by *eLife*. Your article has been reviewed by two peer reviewers, one of whom is a member of our Board of Reviewing Editors, and the evaluation has been overseen Wenhui Li as the Senior Editor. The reviewers have opted to remain anonymous.

The reviewers have discussed the reviews with one another and the Reviewing Editor has drafted this decision to help you prepare a revised submission.

Summary:

Kino et al. use RNA seq and ribosome profile to investigate mechanisms underlying the control of viral and host mRNA translation in cells infected with Influenza A virus (IAV). Significantly, they elegantly show that the potent suppression of host protein synthesis observed in IAV-infected cells does not result from preferentially IAV mRNA translation. Instead, declining host protein synthesis reflects virus-induced alterations to the composition of the underlying infected cell mRNA pool where viral mRNAs become the predominate species. For the first time, the variable impact of IAV infection on the host mRNA population is revealed and host transcripts able to sustain translation in IAV-infected cells are identified. Host transcripts that were less affected by IAV infection were significantly shorter in length and had higher GC content. Furthermore, IAV infection stimulated translation of host mRNAs responsive to eIF2α phosphorylation, which reduces overall protein synthesis while enhancing translation of specific transcripts whose products support adaptive stress responses. In particular, host mRNAs refractory to IAV-induced host shut off are involved in cell maintenance processes such as oxidative phosphorylation, which the authors show is continuously required throughout the viral replication program for optimal viral genome replication. This work is important to our understanding of virus-host interactions in IAV-infected cells not only because it addresses a fundamental biological problem, but also as it conclusively overturns incorrect ideas in the older literature that proclaimed preferential translation of IAV mRNAs (often without the rigorous evidence to warrant such a conclusion). The experiments are superbly and expertly executed, the data are clearly presented and the manuscript is well written.

Essential revisions:

1) It is desirable to illustrate the occurrence of "host shut off" in their particular system by pulse-labeling of proteins with [^35^]-Met at different times after infection followed by SDS-PAGE.

2) Detection of reads that align to the IAV minus strand RNA is puzzling. The authors speculate that these RNA fragments are not generated by translating ribosomes but are due to the formation of viral nucleoprotein particles. Could they prove this using puromycin (which dissociates polysomes but not RNP)?

3) Goodman et al., 2007 as referenced used mouse cells in their study of eIF2α phosphorylation in IAV-infected cells. This work by Kino et al. utilizes human A549 lung carcinoma cells. Isn't it possible that the capacity of these different cells (human transformed, tumorigenic vs mouse non-transformed, non-tumorigenic MEFs) might vary in their capacity to buffer against accumulation of phospho-eIF2α? The authors should consider blotting for total and phosphor eIF2α in IAV infected A549 cells under the experimental conditions utilized herein to verify that they can or cannot detect changes in phospho-eIF2α abundance.

4) Were any signatures of the integrated stress response or unfolded protein response (UPR) noted among eIF2α phosphorylation-responsive genes? Also --activation of host p58 IPK has been proposed to prevent accumulation of phospho-eIF2α in IAV infected cells. Were any changes in the abundance of the cellular p58-IPK noted in the overall infected cell mRNA population and/or in ribosome-protected fragments? The authors might consider mentioning these examples specifically at some point in the text or discussion regardless of the findings given the attention they have received in the earlier literature

5) In the Results, subsection “Cellular transcripts reduction along IAV injectionis correlated with transcripts’ length and GC content”, the authors conclude that the shorter, more GC rich host mRNAs, are less effected by IAV infection. The authors might wish to include one or two sentences in the Discussion section addressing why this is the case. For example, could this reflect the substrate specificity of endo PA-X or cap-stealing enzymes or some other mechanism?

6) Figure 6. The authors refer to a significant reduction in IAV "titers." The term "titer" usually refers to infectious virus units – here genome copy number is being measured. Granted that while genome copy number most certainly has some relationship to infectious virus production and pfu, it is not clear how a 5-fold reduction in genome copy number impacts titer. Probably the simplest solution is to just change the wording here to better reflect what is actually being measured experimentally.

---

## [Author Response]

*Essential revisions:*

*1) It is desirable to illustrate the occurrence of "host shut off" in their particular system by pulse-labeling of proteins with [^35^S]-Met at different times after infection followed by SDS-PAGE.*

As was suggested by the reviewers, we now performed metabolic labeling along IAV infection and we establish that our system recapitulates previous observations about host-induced shutoff during IAV infection. These experiments demonstrate that similar to previous reports, IAV infection results in massive production of virus polypeptides from 4 hpi and this is accompanied by reduced background in cellular protein synthesis (Figure 3—figure supplement 1)

*2) Detection of reads that align to the IAV minus strand RNA is puzzling. The authors speculate that these RNA fragments are not generated by translating ribosomes but are due to the formation of viral nucleoprotein particles. Could they prove this using puromycin (which dissociates polysomes but not RNP)?*

The translation inhibitor, harringtonine, immobilizes ribosomes immediately after translation initiation but does not affect elongating ribosomes (Fresno, M., Jimenez, A. & Vazquez, D. Eur J Biochem 72, 1977). Therefore, a 5-minute pre-treatment with harringtonine leads to strong accumulation of ribosomes at translation initiation sites but also generate the depletion of ribosomes over the body of the message (Ingolia, N. T., Lareau, L. F. & Weissman, J. S. Cell, 2011). This property of polysome run off, which is similar in concept to using puromycin, makes it possible to use the harringtonine pre-treated samples to assess if the reads we mapped from the IAV minus strand originate from translating ribosomes.

As expected 5 min pre-treatment with harringtonine led to a strong accumulation of ribosome protected fragments at the first AUG and to depletion of ribosome density from the body of viral mRNAs (Figure 2). In contrast, the protected fragments that mapped to the IAV minus strand were not affected by harringtonine pre-treatment. This point is now illustrated for individual gene (Figure 2, light blue) and in a new metagene analysis that demonstrates no difference in the protected fragments distribution on the viral minus strand following pre-treatment with harringtonine (Figure 2—figure supplement 1). This data together with the analysis of the size distribution of the protected fragments (FLOSS analysis/ Figure 2) strongly suggests that the protected fragments originating from minus strand viral RNA are not generated by ribosome protection.

*3) Goodman et al., 2007 as referenced used mouse cells in their study of eIF2α phosphorylation in IAV-infected cells. This work by Kino et al. utilizes human A549 lung carcinoma cells. Isn't it possible that the capacity of these different cells (human transformed, tumorigenic vs mouse non-transformed, non-tumorigenic MEFs) might vary in their capacity to buffer against accumulation of phospho-eIF2α? The authors should consider blotting for total and phosphor eIF2α in IAV infected A549 cells under the experimental conditions utilized herein to verify that they can or cannot detect changes in phospho-eIF2α abundance.*

As was requested by the reviewers we now added western blot analysis demonstrating that eIF2α is phosphorylated at 4hpi but that this phosphorylation is transient and at 8hpi it is drastically reduced. These measurements align well with our translation efficiency measurements and supports the notion that IAV blocks eIF2α phosphorylation.

*4) Were any signatures of the integrated stress response or unfolded protein response (UPR) noted among eIF2α phosphorylation-responsive genes? Also --activation of host p58 IPK has been proposed to prevent accumulation of phospho-eIF2α in IAV infected cells. Were any changes in the abundance of the cellular p58-IPK noted in the overall infected cell mRNA population and/or in ribosome-protected fragments? The authors might consider mentioning these examples specifically at some point in the text or discussion regardless of the findings given the attention they have received in the earlier literature*

P58-IPK is a cellular inhibitor of PKR and previous studies argued that it is activated at the post-translational level in response to influenza virus infection (Lee, T. G., Tomita, J., Hovanessian, A. G. & Katze, M. G., J. Biol. Chem. 1992) and that its activation is responsible for the inhibition in eIF2α phosphorylation during influenza infection (Goodman, A. G. et al., J. Virol. 2007). Our new western blot analysis supports the observation that influenza virus blocks eIF2α phosphorylation (Figure 4). In our ribosome profiling measurements P58-IPK (DNAJC3) expression levels are reduced by 3-fold, but since our measurements are reporting on the level of translation (and not the levels of the protein) and previous reports suggested that P58-IPK activity is increased but not its expression, these results by themselves do not contradict or support the proposed model. We now discuss these points in our revised manuscript (third paragraph of Discussion).

With regards to general UPR induction, although there is significant overlap between genes that were upregulated at 4hpi and two recent publications that mapped genes that are up regulated following induction of eIF2α phosphorylation (Andreev, D. E. et al., elife 2015 and Sidrauski, C., McGeachy, A. M., Ingolia, N. T. & Walter, P. elife 4 2015), we do not see specific enrichment in UPR genes in the genes that were translationally induced after infection.

5) In the Results, subsection “Cellular transcripts reduction along IAV injectionis correlated with transcripts’ length and GC content”, the authors conclude that the shorter, more GC rich host mRNAs, are less effected by IAV infection. The authors might wish to include one or two sentences in the Discussion section addressing why this is the case. For example, could this reflect the substrate specificity of endo PA-X or cap-stealing enzymes or some other mechanism?

We now discuss these points in the Discussion:

“The magnitude of reduction could only be partially explained by cytoplasmic half-life and interestingly the extent of reduction was significantly correlated with transcript’s length and GC content. […] This notion is consistent with existing In-vitro data showing PA-X preference to ssRNA (Bavagnoli et al., 2015) and with the findings that PA-X lacks obvious sequence or location specificity (Khaperskyy et al., 2016).”

6) Figure 6. The authors refer to a significant reduction in IAV "titers." The term "titer" usually refers to infectious virus units – here genome copy number is being measured. Granted that while genome copy number most certainly has some relationship to infectious virus production and pfu, it is not clear how a 5-fold reduction in genome copy number impacts titer. Probably the simplest solution is to just change the wording here to better reflect what is actually being measured experimentally.

We thank the reviewer for noticing this error. We revised the text and for completeness we added new data from plaque assay that shows we also obtained reduction in viral titers (Figure 6).